# Cryo-EM structures from sub-nl volumes using pin-printing and jet vitrification

Raimond B. G. Ravelli [1✉], Frank J. T. Nijpels[1,2], Rene J. M. Henderikx[1,2], Giulia Weissenberger[1,2], Sanne Thewessem [3], Abril Gijsbers[1], Bart W. A. M. M. Beulen [2,3], Carmen López-Iglesias[1] & Peter J. Peters [1,2✉]

The increasing demand for cryo-electron microscopy (cryo-EM) reveals drawbacks in current sample preparation protocols, such as sample waste and lack of reproducibility. Here, we present several technical developments that provide efficient sample preparation for cryo-EM studies. Pin printing substantially reduces sample waste by depositing only a sub-nanoliter volume of sample on the carrier surface. Sample evaporation is mitigated by dewpoint control feedback loops. The deposited sample is vitrified by jets of cryogen followed by submersion into a cryogen bath. Because the cryogen jets cool the sample from the center, premounted autogrids can be used and loaded directly into automated cryo-EMs. We integrated these steps into a single device, named VitroJet. The device's performance was validated by resolving four standard proteins (apoferritin, GroEL, worm hemoglobin, beta-galactosidase) to ~3 Å resolution using a 200-kV electron microscope. The VitroJet offers a promising solution for improved automated sample preparation in cryo-EM studies.

[1] The Maastricht Multimodal Molecular Imaging Institute (M4i), Division of Nanoscopy, Maastricht University, Maastricht, Netherlands. [2] CryoSol-World, Maastricht, Netherlands. [3] Instrument Development, Engineering and Evaluation (IDEE), Maastricht University, Maastricht, Netherlands. ✉email: rbg.ravelli@maastrichtuniversity.nl; pj.peters@maastrichtuniversity.nl

Within just a few years, single-particle cryo-electron microscopy (cryo-EM) has become a powerful mainstream technique to resolve high-resolution 3D structures of macromolecules. Technological breakthroughs in microscopes, detectors, and processing have contributed to the resolution revolution[1]. However, numerous advancements are yet to be made[2,3]. Reproducible sample preparation has emerged as one of the main bottlenecks[4]. The sample preparation technique for cryo-EM, pioneered by Dubochet and others[5–8], allows particles to be preserved in a thin vitreous layer in their native state. The current leading commercial sample preparation devices still rely on the work of these pioneers. Sample preparation begins with an EM grid that is plasma-treated in a separate instrument to increase hydrophilicity. As this effect is only temporary, the wetting properties depend on the time between glow discharge and sample deposition, temperature, humidity, and cleanliness of the environment[4,9,10]. Next, a few microliters of sample are manually pipetted onto the grid, and >99.99% of it is blotted away by filter paper, after which an aqueous film is allowed to thin by evaporation. During this dynamic process, air–water interfaces will form[5,11,12], which can be detrimental to the structure of interest as proteins tend to adsorb to such interfaces and (partly) denature[11]. The resulting film of sample on a perforated carrier[13–15] has a generally concave shape[11,16] due to drying and draining, with the center being the thinnest. The grid is held by tweezers and plunged into a bath of cryogen to vitrify the sample[17], so that it can be observed in the vacuum of the microscope. This vitrification should be fast enough to prevent the formation of ice crystals. Grids are stored under cryogenic conditions until analysis. For microscope loading and unloading, the fragile grid needs to be assembled into either a sturdy cartridge or the tip of a holder: both processes are cumbersome and often harmful to the fragile grid.

Many steps of the above protocol require skilled operators, careful handling, clean tools, clean cryogens, and controlled environments. Some steps are intrinsically difficult to control accurately, such as positioning of the filter papers, blotting force, humidity, and flatness of the filter papers, as well as the position and shape of the tweezers. As a result, reproducibility is lacking. The required training and skills to obtain reliable grid quality is a significant entry barrier for new scientists. The increasing demand and evolution of cryo-EM call for improved, scalable methods. We have developed a method which allows for better control and, ultimately, minimal operator intervention. Our method consists of (i) an integrated glow-discharge module to control and minimize the time between plasma cleaning and sample deposition; (ii) pin-printing for sample application, which requires only sub-nanoliter sample volumes and eliminates sample blotting; and (iii) jet vitrification, which allows for the handling of autogrids. We integrated these features into a single setup, termed the VitroJet, and used it to prepare four standard proteins to obtain high-resolution single-particle reconstructions.

## Results

**Sample carriers**. All results below were obtained with pre-mounted autogrids[18,19]. Autogrids were initially developed for increased robustness in order to allow automated handling of grids in cryo-transmission electron microscopes (TEMs), such as the Titan Krios, Glacios, and Arctica (Thermo Fisher Scientific). Traditional vitrification devices are not able to vitrify pre-mounted (assembled at room temperature) autogrids. Therefore, autogrids are normally manually assembled under cryogenic conditions by clamping an EM grid into a sturdy cartridge by means of a flexible C-clip spring (so called post-mounting). The jet vitrification procedure described below overcomes this limitation.

**Glow discharge module**. Traditional vitrification devices use external glow discharge modules. We characterized this procedure by determining contact angles for typical glow discharge settings used in our lab. An ELMO Glow Discharge System (Cordouan Technologies) was operated at 7 mA, 0.35 mbar, and 30 s glow discharge time. Contact angles (measured using a Krüss drop shape analyzer, model DSA25) of $23 \pm 7°$ ($n = 6$) were obtained for minimal transfer time (1½ min) between the glow discharge system and the drop shape analyser.

Within the VitroJet, the glow discharge module is an integral part of the device. Each grid is glow discharged separately, and transferred within 10 s from the glow discharge unit into the process chamber. We evaluated different glow discharge settings (0.3–20 mA; 0.05–0.2 mbar), using different grids (Quantifoil R2/2 Cu300, UltraAuFoil R2/2 Au300), different treatment times (5–40 s), and different durations between the glow discharge treatment and the contact angle measurement (1–17 min). Contact angles of $17 \pm 3°$ ($n = 5$) could be obtained from grids glow-discharged in the VitroJet. In general, longer glow discharge treatment gave lower contact angles. Not surprisingly[20], longer delays (17 min) between glow discharge and contact angle measurements gave larger contact angles (35°). The glow-discharged grids did not show any visible damage. The single particle data described below were obtained using grids that were glow discharged for 30 s at 0.5 mA and 0.1 mbar.

**Process chamber**. In order to minimize sample evaporation, current vitrification devices such as the Vitrobot, Leica EM GP, and the Gatan Cryoplunge apply the sample inside a humidified chamber. Passmore[4] recommended to keep the relative humidity (RH) surrounding the specimen support at 100% to prevent changes in the solute concentration prior to freezing. However, it is difficult to achieve a reliable 100% RH. Once the air within the chamber is fully saturated with water vapor, condensation will occur. Water droplets on the humidity sensor and/or the grid compromise the reproducibility of the experiment: humidity sensors only work reliably under non-condensing conditions.

We first studied the theoretical and reported evaporation rates of thin layers of water on grids. Different evaporation velocities of thin (suitable for TEM) water layers have been reported, depending on temperature and RH: 5 nm/s at 4 °C and 90% RH[4] and 40–50 nm/s at 20 °C and 40% RH[12]. We modeled the evaporation of a thin layer of water on a grid (Supplementary Note 1). Parameters within this model include partial water pressure in air, saturated vapour pressure, velocity of the air flow over the sample, thermal conductivity of the sample carrier, and shape of the deposited water layer. For a given temperature and RH, the dew-point temperature of the grid can be calculated (Fig. 1a). At <100% RH, the dew-point temperature of the grid will be lower than the temperature in the chamber. Our model predicts that the layer thickness decreases by 70 nm/s for every °C of dew-point error, for a thin line of pure water deposited onto the grid at 93% RH in the chamber (Fig. 1b; Supplementary Note 1).

In order to prevent sample evaporation, the sample itself thus needs to be maintained at dew point. Based on the relationship shown in Fig. 1a, we built a fast feedback loop to maintain the grid at dew-point temperature. The feedback loop, implemented in LabView, monitors the RH and temperature of the process chamber and calculates the dew-point temperature at a frequency of 100 Hz. Within the VitroJet, this chamber is maintained at 93% RH. The grid is peltier-cooled through the autogrid cartridge. The optimal grid temperature was offset from its calculated dew point to account for heat exchange between the grid and its environment, resulting in an elevated temperature of the center of the grid compared with the sensor positioned just below the

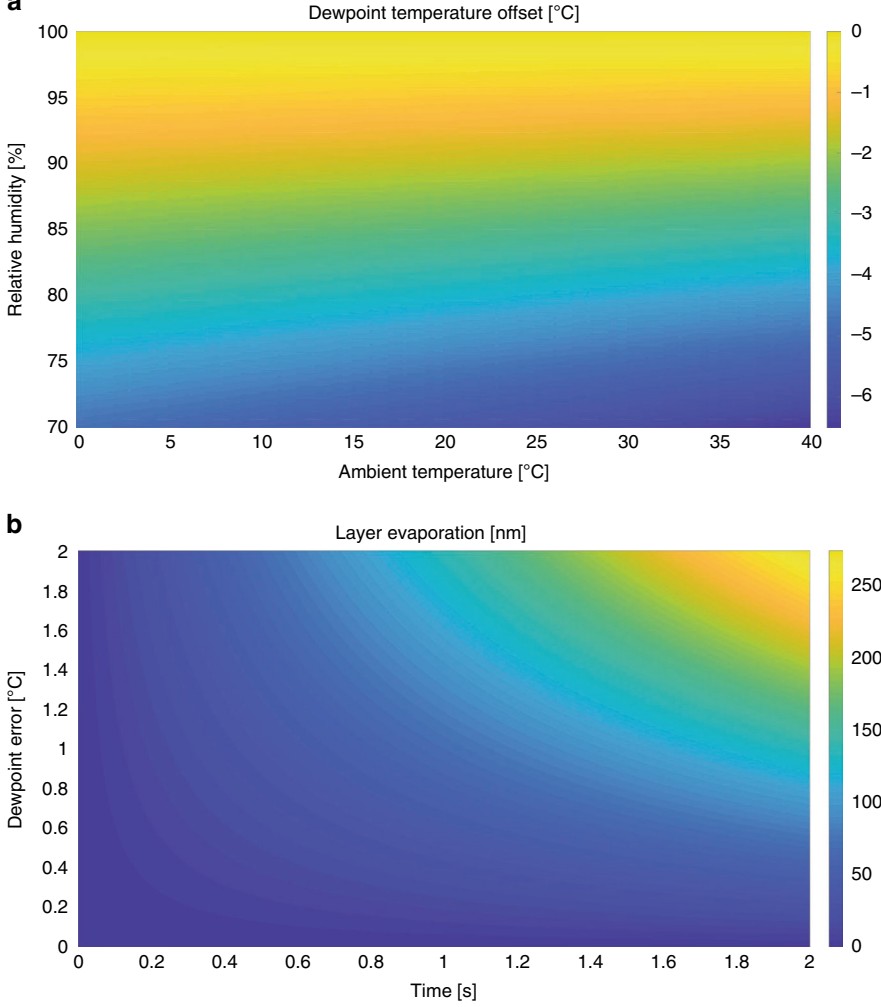

**Fig. 1 Dew-point temperature calculations and predictions. a** Dew-point temperature offset (color scale, in degrees) as a function of the relative humidity and ambient temperature. E.g., when working at 20 °C and 94% relative humidity, the dew-point temperature would be 1 °C (orange) below the ambient temperature. **b** Predicted decrease in thickness of a thin sample layer due to evaporation (color scale, in nm/s) with respect to the time and the dew-point error, which is the difference between sample temperature and its dew point. E.g., for a temperature of 1 °C above dew point, 70 nm of the layer is estimated to evaporate within 1 s.

grid. Even if the edge of the grid would be held perfectly at the dew-point temperature, the estimated temperature of the center can be a few tenths of a degree higher (Supplementary Note 1). Our theoretical model predicts that, if such temperature difference would not be accounted for, the layer thickness could still decrease by 14 nm/s. An incorporated camera enables visual inspection of evaporation and/or condensation on the grid and can be used to finetune the dew-point feedback loop (Supplementary Movies 1 and 2).

**Pin printing**. To improve sample deposition on EM grids, several developments have been presented, including droplet-based methods[21–25], e.g., on nanowire grids[26], and using capillaries followed by sample thinning by controlled water evaporation[27]. We sought a way to obtain sample thickness layers suitable for cryo-EM without the need for blotting, nanowire grids, and/or extra water evaporation steps. In our pin-printing method, a sub-nl volume of sample is deposited onto the grid using a pin. In order to determine the relative positions of the pin and the grid, we position them subsequently in the focal plane of the camera. Using these calibrated positions, the pin can be moved to a defined distance of the grid.

A stock sample volume of 0.5 μl is introduced into the process chamber by a positive displacement pipette. A metal pin is dipped into the sample to collect a sub-nanoliter volume. The pin is cooled down to dew point to prevent evaporation of the tiny droplet at its tip. The pin is moved to a predefined distance (typically 10 μm) from the carrier surface such that the sample forms a capillary bridge between the pin and grid. Once this bridge is formed, the pin is moved along the surface of the carrier). Capillary forces ensure that the liquid bridge follows the pin. If this parallel movement is sufficiently fast, a thin film will be deposited due to the viscous shearing on the liquid. The film thickness $h$ is determined by the relative speed of the pin $u$ moving over the carrier surface, the stand-off distance $\delta$ between pin and carrier, the viscosity $\mu$, surface tension $\sigma$, and surface properties of the pin and the carrier (Fig. 2a). For the data shown here, we used a velocity of 0.3 mm/s (Supplementary Movie 3). Moving at a higher speed should result in thicker sample layers and vice versa[20] whereas using a larger stand-off distance should result in a thicker layer. The pin diameter itself will influence the range of stand-off distances one can use and the area one can write. In summary, the pin-printing technique opens up a variety

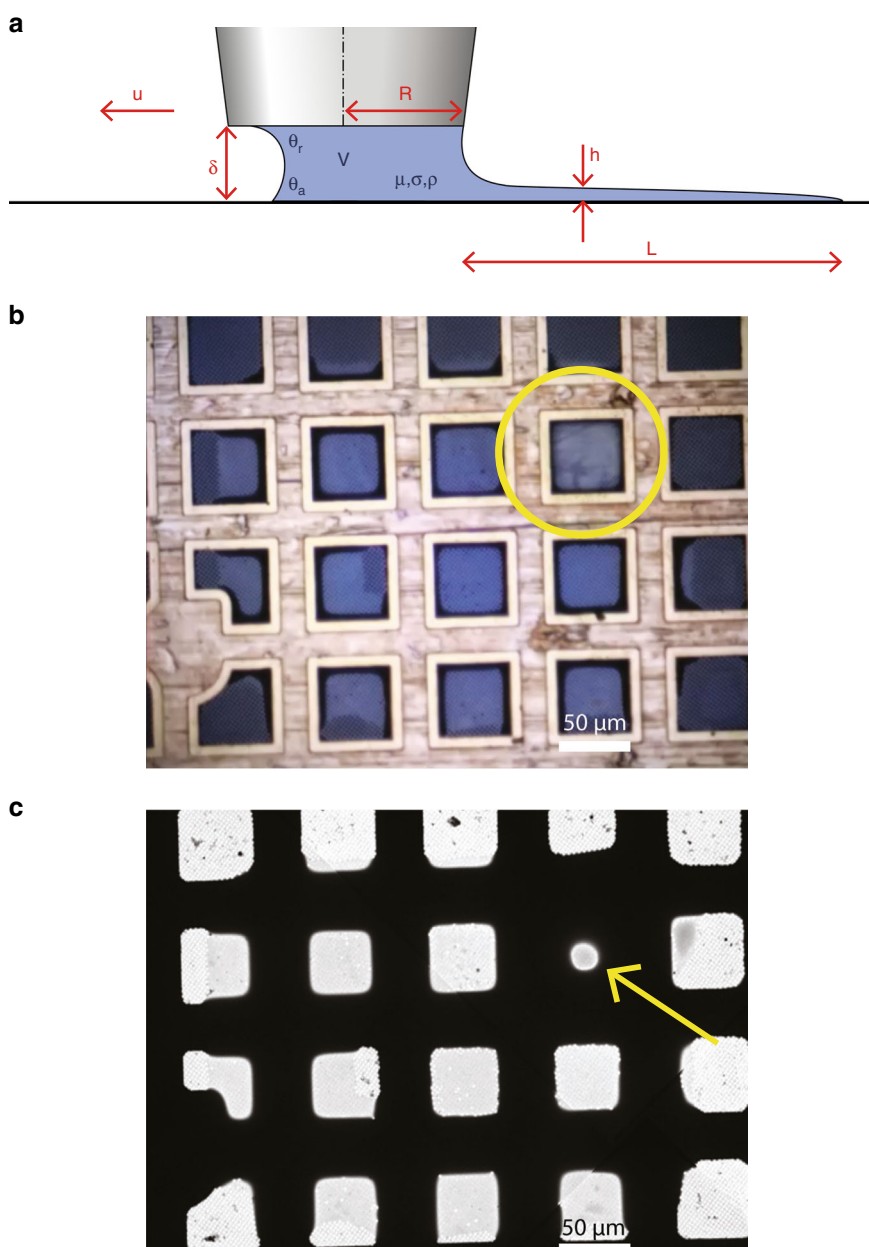

**Fig. 2 Pin-printing parameters and resulting sample deposition. a** Schematic overview of relevant parameters for pin printing. The deposited layer in height ($h$) and length ($L$) is a result of the sample properties (viscosity $\mu$, surface tension $\sigma$, density $\rho$), contact angles (advancing at the grid $\theta_a$, receding at the pin $\theta_r$), and writing parameters (standoff distance $\delta$, pin radius $R$, pin velocity $u$). **b** Photograph of the grid by the integrated camera shows the sample applied by pin printing. The pin is indicated by a yellow circle. **c** Low magnification cryo-EM atlas of the deposited and vitrified sample. One thicker square (arrow) is the result of retraction of the pin from the grid. Scale bars represent 50 μm.

of possibilities to optimize sample layer thickness and counteract variations in fluid properties.

While printing could be performed in any pattern, we found that printing in a partly overlapping spiral pattern (Fig. 2b, c) resulted in a more uniform thickness compared with, for example, in one straight line across the carrier. Finally, the pin retracts from the surface and the remaining volume in the liquid bridge breaks up, leaving a thicker droplet at the end of the deposited line. Because only a sub-nanoliter volume is deposited per printing, numerous grids can be printed from the same stock volume of 0.5 μl.

Three factors could alter the film layer thickness between deposition and vitrification. These include liquid flow, evaporation, and condensation. Liquid flow is characterized by the redistribution time scale $T_r$ for a surface-tension-driven thin-film

flow[28]. $T_r = \mu d^4/\sigma h^3$, and is primarily dependent on $d$, the distance the fluid has to travel, and the thickness $h$ of the sample layer. Redistribution occurs on two timescales: the first is very fast for thick layers of sample adhering to the grid bars, whereas the second is a much slower redistribution of liquid within the thin layer over the grid square surface. When the pin moves over a grid square, wicking by the grid bars takes place within milliseconds, resulting in an unusually thick layer just around the edges of the grid square which cannot be prevented in the deposition process. The second liquid redistribution is slower. For example, for a 50 nm thin layer of water ($h$), and $\mu \approx 10^{-3}$ Pas being the dynamic viscosity of the liquid, and $\sigma \approx 0.07$/m being the surface tension of the liquid, the redistribution time scale would be >10 min.

Figure 2c shows a low magnification EM overview of the pin-printed area, whereas Supplementary Fig. 1 shows the ice distribution within a grid. Both figures show that the ice layer is somewhat thicker at the start of pin printing (middle of the atlas) compared with the end (outside of the squared spiral). Before the start of deposition, the pin approaches the grid and squeezes the droplet, leaving an excess of sample in the center. This increased thickness is maintained due to time scale of liquid redistribution. Alteration of the deposited sample layer could still occur by evaporation or condensation. Both effects are mitigated by a tight control of the dew-point feedback loop described above. Overall, the sample layer thickness obtained directly after pin printing will be very similar to the vitrified one.

**Vitrification module**. After sample deposition, traditional vitrification devices rapidly plunge the bare grid into a bath of cryogenic liquid to vitrify the sample. Plunge vitrification starts cooling down from the bottom of the grid upward, and boiling of the coolant can form a gaseous insulating layer at the surface of the grid[17]. This process can compromise the cooling time of the sample, which is estimated to be $10^{-4}$ s in the most favorable case of thin-layer vitrification[7]. These current devices are not able to vitrify pre-mounted autogrids due to the extra thermal mass of the cartridge. The sturdy rim of autogrids would hit the cryogenic liquid first, whereas the area of interest (the sample on the middle of the grid) is cooled later at speeds too slow to prevent ice-crystal nucleation.

Inspired by cryofixation of thick tissues for room-temperature ultrastructural studies[29], we devised an alternative way to vitrify samples in pre-mounted autogrids: jet vitrification. Two streams of cryogenic liquid hit the sample and its carrier in the center (Supplementary Movie 4), cooling it down to temperatures below 130 K in <1 ms. As jet vitrification cools the autogrid from the center outward, the rim of the cartridge is cooled down last. The jets continues to spray for 50 ms to precool the cartridge; hereafter, the autogrid is submersed into a bath of liquid ethane to fully cool down the cartridge rim and gripper. The grid is subsequently slowly moved out of the ethane bath to allow excess ethane to flow off and prevent ethane solidification. Following vitrification, the gripper transfers the autogrid to a spring-loaded storage container in liquid nitrogen to connect directly to the microscope-loading workflow.

The preparation for jet vitrification is similar as for the Vitrobot. A cryochamber is precooled by liquid nitrogen before introducing the vitrification cryogen. At the start of the process ~13 ml of ethane or other cryogen (compared with the 6–7 ml used standard for a Vitrobot) is condensed in a precooled bath surrounded by a liquid nitrogen chamber. As the jetting requires <100 ms in total with ethane returning to the cryogen bath, ethane is recycled for the grids. Following the preparation of 8 grids, the ethane is typically refilled to maintain uniform vitrification conditions.

We conducted experiments with different cooling media at different temperatures, including liquid nitrogen (77 K), ethane/propane (37%/63% (v/v) at 79 and 93 K), and ethane (99 K). Of these different media, liquid nitrogen and liquid ethane/propane cooled the grid to the lowest final temperature, but the highest cooling rates were obtained with liquid ethane (Supplementary Fig. 2). These cooling rates were measured with 25 μm constantan wires (Bare thermocouple wire, Omega, Norwalk) woven into a copper grid mesh to form a thermocouple: such wires are expected to cool much slower than the very thin sample. The center of the grid showed significantly higher cooling rates with jetting compared with plunging, both for autogrids as well as regular grids (Supplementary Fig. 2).

**Integration**. The above steps (glow discharge, process chamber, pin-printing, and jet vitrification) were implemented into the VitroJet (Fig. 3a). A supply cassette provides up to 12 pre-clipped autogrids. A gripper picks up each autogrid individually and transports it sequentially through each of the different steps for sample preparation (Fig. 3b; Supplementary Movie 5). The gripper is dried by nitrogen gas within the glow discharge unit before picking up the next carrier.

The cycle time of the workflow is ~3 min. Most of this time is taken up by the glow discharge module: evacuation of this chamber takes ~20 s (twice per cycle), while the glow discharge itself takes 30 s. For pin printing, calibration of the sample carrier and pin position within the process chamber requires <30 s, and the sample deposition a few seconds. Transfer of the grid between the process chamber and the vitrification chamber is completed within 80 ms. Vitrification of the sample lasts <1 ms, whereas another 300 ms is used to deeply cool down the entire autogrid assembly. Grid removal from the ethane bath and transfer requires <30 s.

Before each use, the VitroJet undergoes several automatic preparatory tasks, which are completed in ~15 min and includes cooling down the vitrification unit, filling the liquid ethane reservoir, and equilibrating the process module at the right humidity. The temperature-controlled liquid ethane reservoir is cooled by a liquid nitrogen bath.

**Structure determination by single-particle analysis (SPA)**. To validate the VitroJet, we prepared samples of several standard proteins (apoferritin, GroEL, worm hemoglobin, beta-galactosidase) and performed high-resolution SPA (see Methods). Each sample was pin-printed on ~16 squares of 300-mesh grids with perforated foils (R1.2/R1.3) and jet-vitrified using liquid ethane. Atlas overviews collected within the cryo-EM show excellent correlation with the visualization of the deposition within the VitroJet just prior to vitrification (Fig. 2b, c). Cryo-EM data were collected on a 200-kV FEI Arctica microscope. From the squares that were pin-printed, most holes could be selected for data collection. Holes close to the grid bar were skipped to avoid thicker ice due to the wicking of the grid bars. Micrographs were recorded between 0.5 and 2.0 μm underfocus, showing good contrast (Fig. 4).

The data, processed with Relion[30], yielded good 2D classes. After iterations both with particle picking and extraction, (local) contrast transfer function estimation, and 2D and 3D classifications, 3D reconstructions of ~3-Å resolution were obtained for each of these four proteins (Fig. 4). Apoferritin gave a 3D reconstruction of 2.49 Å using 47,209 particles (O symmetry). For GroEL, we obtained a 3D reconstruction of 2.94 Å using 9809 particles (D7 symmetry). 3D reconstructions of worm hemoglobin and beta-galactosidase had respective resolutions of 3.11 Å (using 10,488 particles, D6 symmetry) and 3.11 Å (using 15,252 particles, C7 symmetry). Models refined against the four reconstructions were in accordance with earlier published models demonstrating the success of the VitroJet in preparing suitable samples for cryo-EM studies.

## Discussion

Traditional cryo-EM sample preparation methods require multiple manual steps and depend on ill-controlled parameters such as blotting force, grid positioning, and time between glow discharge and sample application. Here, we present a workflow that minimizes operator dependency and provides control over relevant parameters. Prior to starting the sample preparation cycle, parameters regarding glow discharge, dew point, pin printing and jetting can be set. After initiation, the process is executed in an

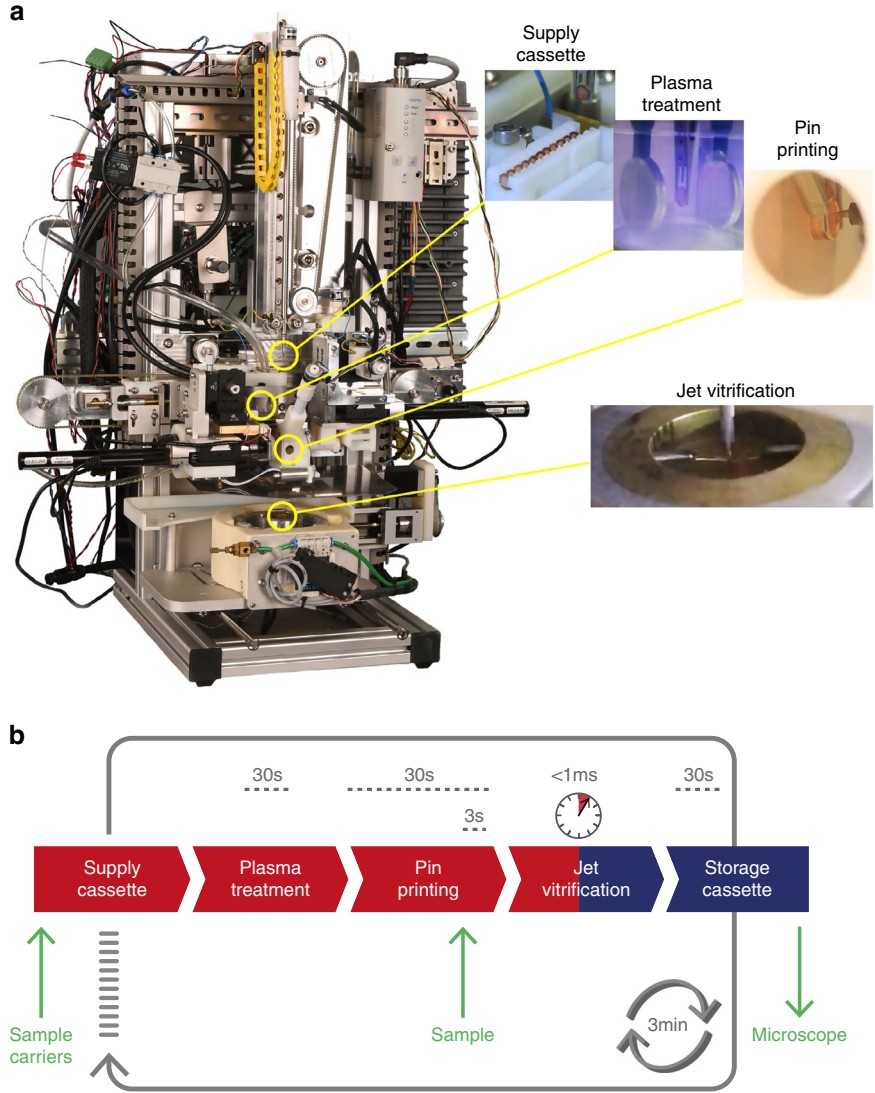

**Fig. 3 The VitroJet workflow. a** The device and **b** the workflow. Sample carriers (up to 12) are introduced in a supply cassette and sample (up to 0.5 μl) in a pipette tip (yellow arrows). The VitroJet automatically processes the cassette by sequentially passing sample carriers through the plasma treatment, sample deposition by pin printing, jet vitrification and finally storing the grid in a cryogenic cassette ready to enter the electron microscope (green arrow). The supply cassette is at room temperature (red), the storage cassette at cryogenic temperature (blue). An animation showing all subsequent steps is shown in Supplementary Movie 5.

automated fashion. In an era where automation has made so much impact on microscope alignment, data collection, processing, and model building, automated control over sample preparation is a mandatory next step.

Samples applied to pre-mounted autogrids are difficult to blot and even more difficult to vitrify using the existing leading commercial devices or even the next-generation blotless commercial device Chameleon[31]. We overcame both problems by using pin printing, which does not require blotting, and jet vitrification, which yields superior cooling rates starting from the center of the grid where the sample is located. In addition to circumventing the problems associated with blotting, pin printing requires minute volumes of sample, which enables the study of macromolecules for which only micrograms can be obtained.

Proteins tend to absorb to the air–water interface where they can denature[11,32,33]. It seems intuitive that reducing the time the sample is exposed to such an interface would help to prevent protein denaturation[16]. However, using the Stokes–Einstein equation, it has been calculated that even for a minimal residence

time of ~1 ms, particle-surface interactions will occur dozens of times before the water is frozen[34,35]. Unfortunately, all existing devices have residence times varying between 11 ms (ref. [25]) and several seconds and therefore cannot prevent particle–surface interactions. For the VitroJet, the residence time varies between seconds for the first written square down to 80 ms for the last square written prior to vitrification. A promising complementary approach would be to control the surfaces present in the specimen support[34,36], as demonstrated for yeast fatty acid synthase applied to a substrate of hydrophilised graphene[11]. We tested the device with different types of grids (continuous carbon, graphene oxide, different hole sizes, UltrAuFoil) and found the pin printing procedure to be compatible with a multitude of (modified) grid supports.

Jet vitrification was originally demonstrated 40 years ago, on thick tissue spanned over a holder[29]. The method was used as a prelude to freeze substitution, resin embedding and sectioning, for ultrastructural studies performed at room temperature. Sample evaporation was not a concern for the bulky tissues used.

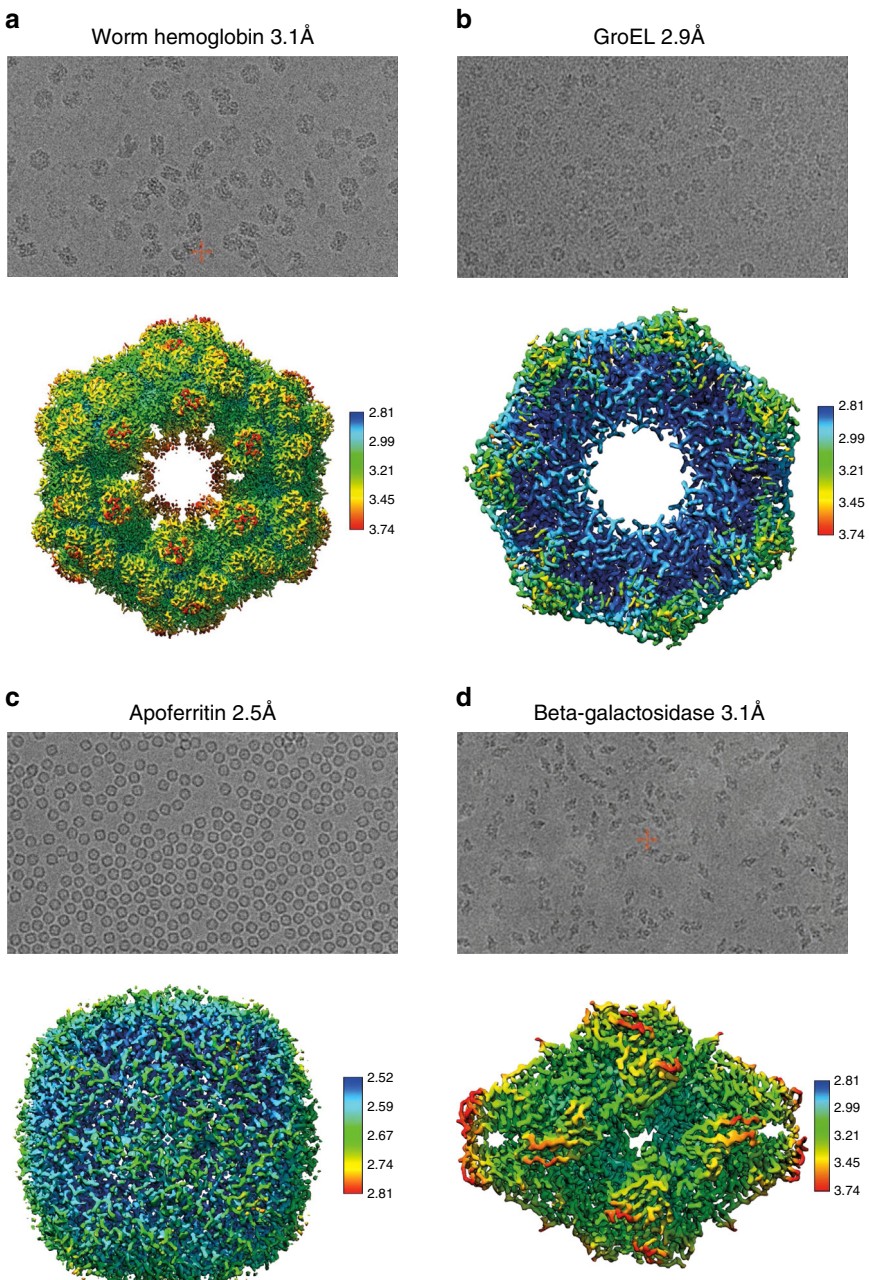

**Fig. 4 Sample preparation of four standard proteins by the VitroJet, and validated by cryo-EM analysis.** Micrographs (top) and reconstructions (bottom) of worm hemoglobin, GroEL, apoferritin, and beta-galactosidase. The reconstructions are colored according to local resolution as calculated using Relion.

Here, we adapted jet vitrification to obtain thin layers of macromolecular samples. One might expect that sample evaporation problems would be insurmountable and that the jets would blow away the thin liquid layers of macromolecules prior to vitrification, resulting in empty holes. However, we have demonstrated the opposite. We believe that the sample already vitrifies before the liquid cryogen hits the sample[37].

The setup of the VitroJet is modular, making it easier to incorporate future developments to further advance the cryo-EM field. The pin printing applies sample to a part of the available area of the grid. Future generations of the VitroJet could be equipped with multiple pins moving simultaneously over different parts of the grids. Such schemes would enable higher-throughput screening as well as time-resolved studies[38,39], e.g.,

combined with laser excitation. The process chamber provides the ability to control condensation as well as evaporation for an extended duration of time on a specific layer thickness, which could also offer benefits for soft condensed matter studies. While the VitroJet described here was developed for SPA, we aim to develop a branch of the VitroJet dedicated to the preparation and vitrification of cellular samples. Vitrification of cells is inherently more difficult than that of purified macromolecular samples. For example, it was stated that the center of HeLa cells clearly undergoes incomplete vitrification[40]. Preliminary results indicate that jet vitrification will help to reduce this problem, which would be a true asset for in situ structural biology. The VitroJet offers much-needed innovations in sample preparation, which will accelerate and perhaps even revolutionize future cryo-EM studies.

## Methods

**Protein purification**. Human apoferritin overexpressed in *E. coli* was kindly provided via Evgeniya Pechnikova (Thermo-Fisher) by Dr. Fei Sun (Institute of Biophysics, Chinese Academy of Sciences). Chaperonin-60 from *E. coli* (GroEL) was ordered from Sigma (C7688), dissolved in a buffer comprised 50 mM Tris (pH 8.0), 100 mM KCl, 10 mM MgCl$_2$, 2 mM DTT, and 80 mM trehalose, and used without further purification at a concentration of 10 mg/ml. Approximately 400-μl blood from earthworm *Lumbricus terrestris* (WHBG) was extracted from the seventh segment of the body[41] and injected into a Superose® 6 Increase 10/300 GL size-exclusion chromatography column (SEC) with an elution buffer of 20 mM Tris (pH 8.0), 150 mM NaCl, and 10 mM CaCl$_2$. After purification, WHBG was concentrated to 6 mg/ml. Beta-galactosidase (β-gal) from *E. coli*, ordered from Sigma (G5635), was further purify by SEC with a Superdex® 200 Increase 10/300 GL column and eluted with 25 mM Tris (pH 8.0), 50 mM NaCl, 2 mM MgCl$_2$, and 1 mM TCEP. The ligand phenylethyl β-D-thiogalactopyranoside (PETG) was purchased from Sigma-Aldrich (catalog #P1692) and prepared as described[42]. We used a final β-gal protein concentration of 4 mg/ml with 7.5 mM PETG.

**Sample preparation**. Quantifoil R1.2/1.3 Au300 and UltraAufoil R1.2/1.3 Au300 grids (Quantifoil Micro Tools, Jena, Germany) were used as sample carriers. Grids were pre-clipped before entering the VitroJet. Sample deposition occurred in a climate chamber at room temperature with a humidity of 93%, where pin and grid were cooled toward dew-point temperature. A sub-nanoliter volume of sample was pin-printed onto the grids with a writing speed of 0.3 mm/s. Samples were vitrified within 80 ms after sample deposition by two pressurized jets of liquid ethane for 50 ms at 99 K and 1 bar.

**Single-particle cryo-EM**. Micrographs were collected on a 200-kV Thermo Fisher Tecnai Arctica microscope equipped with a Falcon3 detector. For each standard protein, micrographs were collected with a calibrated pixel size of 0.935 Å. For apoferritin and GroEL, total integrated electron flux of ~40 e$^-$/Å$^2$ in counting mode at a defocus range of 0.7–1.5-μm underfocus was used. For WHBG and β-gal, a total integrated electron flux of ~43 e$^-$/Å$^2$ in counting mode, at 0.6–1.4-μm underfocus for WHBG and 0.5–1.3-μm underfocus for β-gal, was used. We recorded 383 movies over 48 s for apoferritin (5 squares, 44–76 holes/square), 1284 movies over 69 s for GroEL (8 squares, 67–201 holes/square), 1115 movies over 77 s for WHBG (9 squares, 49–222 holes/square), and 718 movies over 78 s for β-gal (5 squares, 48–224 holes/square).

**Image processing**. The images were processed in Relion[43], where the frames of the movies were aligned and averaged using a Bayesian approach as described[44]. The contrast transfer function (CTF) parameters were calculated with Gctf[45]. Afterward, a subset of micrographs was used to pick ±500 particles manually for initial 2D classification. These 2D classes were used for an iterative, automated particle-picking procedure where both the references and the autopicking parameters were improved using a subset of the micrographs. The complete data set was autopicked, particles were extracted and subjected to an iterative 2D classification scheme to reject bad particles. After 2D classification, 3D auto-refine was performed using starting models based from the PDB (4W1I, 2YNJ, 1X9F, 6CVM). The resulting reconstructions were low pass filtered and projected for another iteration of picking, 2D classification, and 3D auto-refine. Local CTF refinement, and local symmetry in the case of WHGB, resulted in the final maps. Given resolutions were estimated based on established, Fourier Shell Correlation[46] standards[47], as directed by the program Relion.

**Model refinement**. The PDB starting models mentioned above (4W1I, 2YNJ, 1X9F, 6CVM) were superimposed on the sharpened cryo-EM maps. Models were refined iteratively through rounds of manually adjustment in Coot[48], real space refinement in Phenix[49], and structure validation using MolProbity[50].

## Data availability

The source data underlying Supplementary Figs. 1b and 2 are provided as a Source data file. Other data are available from the corresponding authors upon reasonable request.

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

## Acknowledgements

We thank Dr. Fei Sun (Institute of Biophysics, Chinese Academy of Sciences) for providing apoferritin sample, Pascal Huysmans and Paul Kwant (IDEE, Maastricht) for engineering input, Peter Frederik for helpful discussions and Hang Nguyen for critical reading of the paper. Hans Duimel and Hirotoshi Furusho provided technical support from the UM Microscopy Core Lab, Giancarlo Tria helped with initial experiments, Paul van Schayck with the IT infrastructure, and Roger Jeurissen with the theoretical framework. This research received funding from the Netherlands Organization for Scientific Research (NWO) in the framework of the Fund New Chemical Innovations, numbers 731.014.109 and 731.016.407, as well as from the Province of Limburg, the Netherlands.

## Author contributions

P.J.P., C.L.I., and R.B.G.R. designed and directed the project; F.J.T.N. designed and constructed the machine with input from all authors and B.W.A.M.M.B. as project leader; S.T. did preliminary tests; R.J.M.H. and G.W. performed the experiments; A.G. prepared the samples; R.B.G.R., R.J.M.H., and G.W. analyzed the data; R.B.G.R., R.J.M.H., and G.W. wrote the paper with input from all authors.

## Competing interests

The University of Maastricht filed patents with R.B.G.R., F.J.T.N., R.J.M.H., S.T., B.W.A.M.M.B., C.L.I., and P.J.P. as inventors regarding sample preparation for cryo-EM as outlined in this paper. P.J.P. is a shareholder and CSO of the start-up CryoSol-World that holds the licensees of these submitted patents.

## Additional information

**Peer review information** *Nature Communications* thanks John Rubinstein and the other, anonymous, reviewer(s) for their contribution to the peer review this work. Peer reviewer reports are available.

