## [Peer Review File · Nature Communications]

Reviewers' comments:

Reviewer #1 (Remarks to the Author):

This paper describes a novel method for vitrification that has several potential advantages over currently available methods. Some of these include; in process glow discharge for improved control, ethane jet vitrification which enables using pre-clipped grids, pin printing for reducing sample volume, reduced sample evaporation, and overall reproducibility of sample preparation. I am fully in agreement that the device offers a promising solution for vitrification but I feel that the authors should flesh out some of their results and claims as listed below.

- Regarding glow discharge advantages: Their discussion of contact angles makes important claims for reproducibility and control of the in-process method vs. external glow discharge methods. But the quantitative results seem to reduce to a vague mention of 35 degrees vs. 25 degrees contact angle. To get a really thin film across a large surface area contact angles need to be less than 1 degree. So is 35 vs. 25 degrees really significant? What does this mean in practice for the final results?
- In the Results section, under the topic "Glow discharge module" in the second paragraph, it reads "the data below were obtained" but not sure what these data are or where they are to be found.
- The authors discuss at some length the feedback loop and mathematics behind producing a controlled layer of ice thickness. This is indeed a critical point and precise control of ice thickness would be a major boon to the entire field. However, the complicated mathematics they present is then not backed up by quantitative results. For instance: how long does it take to optimize the many parameters responsible for the ultimate ice thickness, h ? Once these are determined, can they be re-used for the next sample? What does it mean to "fine-tune" the feedback loop? What parameters are used in this fine tuning? How reproducible is this? From day to day? Grid to grid? Sample to sample?
- The most critical issue related to the previous point is that the authors do not link the measurement of the thickness of the deposited water layer, h , to the actual measurable thickness of the ice on the subsequently vitrified grid. They mention the thickness of holes in one square (of one sample?) as being consistently $\sim 40\text{nm}$ but they do not provide error bars or other statistics. I think it is quite important to provide estimates of ice thickness for many different squares obtained at both the beginning, middle and end of the (rather long, see below) pin printing procedure. This would help to determine whether there is indeed limited evaporation or if there are other effects occurring during the process. Similarly, it is important to measure ice thickness across multiple samples (and multiple squares) so as to judge the overall reproducibility – and usability – of the device. In figure 4, the authors should also provide multiscale images of the grid to show the consistency of the ice thickness across different squares and across different samples. This is especially important as the authors claim that the device "leaves behind a sample layer of uniform thickness suitable for cryo-EM SPA studies."
- The authors state that: "as EM grids do not have a perfectly flat surface, the absolute reference positions of the pin and autogrid are calibrated separately in the VitroJet using the camera". What does this mean? How long does it take? How easy and reproducible is it? Once the initial contact distance is measured how often do they encounter difficulties during the pin printing itself with bent or wrinkled grids?
- The speed of movement during pin printing is stated to be 0.3 mm/s. So the entire course of the pin printing likely takes on the order of several to many seconds? These numbers should be provided. The authors also completely ignore this time course in their subsequent discussion on the effects of interactions with the air water interface. They discuss the time lapse from the end of the pin printing to the subsequent vitrification (80ms) as if this were the critical time lapse whereas this is a small fraction of the total time the sample has dwelt in a thin liquid film due to the slow pin printing procedure. They also fail to cite or discuss a paper (Nobel et al. Reducing effects of particle adsorption to the air-water interface in cryo-EM. Nature Methods) that proved that time courses on the order of 100ms were indeed useful for reducing interactions at the air-

water interface.

- The authors state: "The initially deposited film layer could differ in thickness from the final one that is vitrified" and then mention possible causes: "evaporation, condensation, and liquid flow". They immediately discard the first two as being unlikely due to the dew point control and then continue on to a somewhat complicated argument as to why the third is also not possible based on their measurements. This leaves unanswered the question as to why the ice thickness changes. And once again emphasizes the critical need for more quantitative measurements of these aspects, as commented already above.
- One of the critical steps in this method is jet vitrification, and the authors do not provide much detail. Flushing through a lot of ethane has implications in terms of costs and safety issues and these should be discussed.

Additional comments:

- In the Introduction, I think instead of "absorb" they mean to say "adsorb"
- The authors should mention the new TTP Labtech device, Chameleon, based on the "Spotiton" technology, when discussing commercial options in the introduction.
- The only image of the overall instrument is insufficient and it is rather challenging to imagine the different sections of the instrument. Perhaps including a block diagram and some magnified images of the different parts of the device would help.
- In Figure 3b, the time taken for making each grid is 3 mins. Is there time involved in refilling the ethane? Is this automated?
- Can the gas mixtures of the glow discharge unit be varied and modified?
- Why are only 16 squares used? How many holes does this include? How many of those holes are useable? To be useful these numbers should be included for all four samples so that reproducibility can be evaluated. I assume that only a single grid was used for each test sample but this should be clearly stated.
- They state: "Holes close to the grid bar were skipped to avoid thicker ice due to the wicking of the grid bars". Perhaps I just missed this in the complex argument about liquid flow but it seems holes next to the grids bars should be thinner not thicker if liquid close to the grids bars is wicked away?
- In Figure 3b, the time taken for making each grid is 3 mins, and it is showed that it is the same for the next grids. How about the time taken to refill ethane? Is it auto-filled? If yes, the authors should include that in this figure.

Reviewer #2 (Remarks to the Author):

This manuscript describes novel and interesting methodological developments to automate cryoEM specimen preparation and make it more reproducible with less operator dependence. In order to accomplish this goal, the authors have created an integrated grid freezing device that includes a glow discharge module, a pin printing sample application module, and jet vitrification module, all of which operate on an already clipped autogrid.

The manuscript is an interesting and valuable addition to the literature and should be published with minor revisions.

1. The authors have incorporated a feedback loop to keep the grids at dewpoint temperature – This objective is accomplished by visual inspection of the evaporation/condensation rate on the grid coupled with thermoelectric cooling/heating of the grid. More detail of how the feedback camera and software work would be helpful and interesting for the reader. Is there evidence that the entire feedback loop (visualization, electronics, heating/cooling) can operate on the same timescale as the evaporation process?

2. Have the authors tested this device with different types of grids (continuous carbon, graphene oxide, holey carbon, different hole sizes) or with different buffers (detergents, glycerol, etc.)? In the discussion, the authors state that 'the printing procedure is compatible with a multitude of grid supports' and it seems it should work with different types of grids. However, the claim that pin printing on graphene should avoid air-water interface interactions does not seem correct: with standard graphene oxide preparation the protein particle adsorbs to the surface while still in bulk solution, thereby avoiding the situation where the protein particle is in a thin film where it can encounter the air-water interface. With the pin printing method the protein would have to adsorb to the graphene after the thin film has already been created. Therefore, in this situation the protein could still encounter the air-water interface before adsorbing.

In this section the authors state that the air-water interface is referred to as the "the deadly touch". First, this sentence doesn't make sense (how can an interface be a touch?). Second, the phrase is certainly not common.

Also, we believe the first calculation of the frequency with which a protein in a thin film solution will encounter an air-water interface was due to Glaeser (e.g. Taylor, K.A., and Glaeser, R.M. (2008). *J. Struct. Biol.* 163, 214-223)

3. One of the major stated advantages of this instrument is its reproducibility/consistency. The authors state that 'From the squares that were pin-printed, most holes could be selected for data collection'. However, it seems like the authors are in a position to include a more quantitative assessment of the grids they produce (ie. number of good holes, ice thickness, etc per holes that are pin-printed). This kind of assessment can be useful – especially as other grid freezing devices are introduced to the market. Furthermore, does this reproducibility vary under different conditions (see question 2).

4. The authors state that the layer thickness can be tuned by altering the deposition parameters, such as the pin speed. This point is extremely important. Is there experimental evidence that the deposition thickness is related to how thick the ice is in the holes after vitrification? If so, this can be very powerful as ice thickness is a major factor when trying to get a sample to high resolution. It would be nice to demonstrate that ice thickness can indeed be controlled by the user.

John L Rubinstein with Justin Di Trani

Please find below our answers to each of the reviewers' comments. Their original comments are shown in red, our answers in black.

Both reviewers appreciate the novelty and interest of our method. Based on their feedback we scrutinised and, where needed, adjusted our claims and statements. The most important claims of the revised manuscript are:

- In the abstract: "We have developed a new method which allows for better control and, ultimately, minimal operator intervention."
- In the pin-printing section: "In summary, the pin-printing technique opens up a variety of possibilities to optimize sample layer thickness and counteract variations in fluid properties."
- In the discussion: "Here we present a workflow that minimises operator dependency and provides control over relevant parameters" and
- "The VitroJet offers much-needed innovations in sample preparation"

The reviewers ask for more details on, among others, reproducibility, which we have tried to address as best as possible. However, we believe that some of the requests of the reviewers go beyond the scope and the claims of this manuscript.

Reviewer #1 (Remarks to the Author):

This paper describes a novel method for vitrification that has several potential advantages over currently available methods. Some of these include; in process glow discharge for improved control, ethane jet vitrification which enables using pre-clipped grids, pin printing for reducing sample volume, reduced sample evaporation, and overall reproducibility of sample preparation. I am fully in agreement that the device offers a promising solution for vitrification but I feel that the authors should flesh out some of their results and claims as listed below.

1. Regarding glow discharge advantages: Their discussion of contact angles makes important claims for reproducibility and control of the in-process method vs. external glow discharge methods. But the quantitative results seem to reduce to a vague mention of 35 degrees vs. 25 degrees contact angle. To get a really thin film across a large surface area contact angles need to be less than 1 degree. So is 35 vs. 25 degrees really significant? What does this mean in practice for the final results?

We now included more precise measurements, both for the external Elmo device ($23 \pm 7^\circ$ $n=6$) and the VitroJet inline glow-discharger ($17.4 \pm 3.1^\circ$ $n=5$). Russo & Passmore also show a contact angle reduction on a gold foil from $82 \pm 8^\circ$ to $27 \pm 6^\circ$ (s.d., $n=6$), and on graphene from : $91 \pm 0.5^\circ$ to $66 \pm 1.3^\circ$ (std. err.) <https://doi.org/10.1016/j.jsb.2015.11.006>, <https://doi.org/10.1038/nmeth.2931>. This suggests that the contact angle does not need to be less than 1 degree in order to get a very thin film.

We now also included a new reference (Houmard M, *et al.* Morphology and natural wettability properties of sol-gel derived TiO₂-SiO₂ composite thin films. *Applied Surface Science* **254**, 1405-1414, 2007) to support our finding that contact angles increase with aging time. The current practice of using external devices is by definition less controlled between grids, days and users compared to using an in-line glow discharge device which treats each grid individually just prior to sample application. We believe that our modest claims about the benefits of being able to

minimize and control the time between glow discharge and sample application, are not overstated.

2. In the Results section, under the topic "Glow discharge module" in the second paragraph, it reads "the data below were obtained" but not sure what these data are or where they are to be found.

The text now reads "The single particle data described below were obtained using grids that were glow discharged for 30 s at 0.5 mA and 0.1 mbar" to clarify which data we are referring to.

3. The authors discuss at some length the feedback loop and mathematics behind producing a controlled layer of ice thickness. This is indeed a critical point and precise control of ice thickness would be a major boom to the entire field. However, the complicated mathematics they present is then not backed up by quantitative results.

The supplementary materials of our manuscript provide a theory of evaporation and argues for the need of dewpoint control. We agree with the referee that more quantitative data could strengthen the mathematical model but this would probably require a dedicated setup in which the sample layer thickness can be measured real-time with nanometer precision. The revised manuscript does include more info on the feedback loop itself.

For instance: how long does it take to optimize the many parameters responsible for the ultimate ice thickness, h ? Once these are determined, can they be re-used for the next sample? What does it mean to "fine-tune" the feedback loop? What parameters are used in this fine tuning? How reproducible is this? From day to day? Grid to grid? Sample to sample?

The fine-tuning of the dewpoint feedback loop mainly entails the determination of the dewpoint temperature using the camera. This takes about 15 minutes and was done after every hard- and software intervention on the functional model (fumo). Without interventions, the dewpoint calibration is stable and can be used for days and weeks. More details about the feedback loop have now been included in the main text and as supplementary movie 1. See also reviewer 2, point 1.

Once the dewpoint control has been calibrated, the other parameters for the ice thickness h can be optimised. The fluidic properties of the samples we used were relatively comparable (if these would be different, we would tune the velocity or stand-off distance). As a consequence, we did not have to alter parameters extensively. We mainly used a square spiral as pattern with a speed of 0.3 mm/s, a stand-off distance of 10 μm , and adjusted these based on visual feedback of the video camera. Reproducibility can be seen in the box plot presented below comparing four grids (see point 4).

4. The most critical issue related to the previous point is that the authors do not link the measurement of the thickness of the deposited water layer, h , to the actual measurable thickness of the ice on the subsequently vitrified grid. They mention the thickness of holes in one square (of one sample?) as being consistently $\sim 40\text{nm}$ but they do not provide error bars or other statistics. I think it is quite important to provide estimates of ice thickness for many different squares obtained at both the beginning, middle and end of the (rather long, see below) pin printing procedure. This would help to determine whether there is indeed limited

evaporation or if there are other effects occurring during the process. Similarly, it is important to measure ice thickness across multiple samples (and multiple squares) so as to judge the overall reproducibility – and usability – of the device. In figure 4, the authors should also provide multiscale images of the grid to show the consistency of the ice thickness across different squares and across different samples. This is especially important as the authors claim that the device “leaves behind a sample layer of uniform thickness suitable for cryo-EM SPA studies.”

The reported thickness of the holes ($\sim 40\text{nm}$) was indeed only determined for a number of holes, using tomography, and has been removed from the revised version. The referee wants to see estimates of ice thickness for multiple squares during the course of pin printing. We now determined these for four grids using the total image intensity as described by Rice et al

<https://doi.org/10.1016/j.jsb.2018.06.007>. For each grid, a number of holes in every grid square was selected for data collection. Based on the intensities, the ice thickness in that hole was calculated and overlaid on the grid atlas, where the colour corresponds to the thickness.

Below an atlas is shown with ice thickness layer distribution for one grid.

The figure above shows that the ice layer is thicker at the start of pin printing (middle of the atlas) compared to the end (outside of the squared spiral). Before the start of deposition, the pin approaches the grid and squeezes the droplet, leaving an excess of sample in the centre. The centre tends to have a somewhat thicker layer compared to the outer square, creating a gradient. The outer squares have a more uniform thickness defined by the capillary coating process.

Comparison of the videos captured during pin printing and the grid overviews collected in the microscope (as shown in figure 2 of the manuscript), does not reveal obvious evaporation. We note that the middle squares were pin printed first and exposed longest to the environment, however, we do not observe these to be drying up. We therefore believe that evaporation was correctly mitigated by the dewpoint feedback loop (see also point 3 and ref 2, point 1).

Below is a figure that shows statistics on layer thickness of four grids.

A boxplot of ice thickness for every grid has been made, which shows reproducibility. It can be seen that the ice thickness was similar between all the grids prepared in this experiment, albeit somewhat thicker compared to the data reported in the manuscript made with an earlier version of the fumo.

The thickness of the ice layer is relatively uniform compared to *e.g.* the VitroJet, where one could have a huge gradient from empty to EM-untransparent squares on one grid. Within the VitroJet, the icelayer is more uniform, however, not identical in every square. We therefore removed the claim "leaves behind a sample layer of uniform thickness suitable for cryo-EM SPA studies" (from section Pin Printing) along with other changes we made to that paragraph to better explain the positioning of the pin relative to the grid (in accordance to point 5 below).

Both figures shown here have now been included within the supplementary data.

5. The authors state that: "as EM grids do not have a perfectly flat surface, the absolute reference positions of the pin and autogrid are calibrated separately in the VitroJet using the camera". What does this mean? How long does it take? How easy and reproducible its it? Once the initial contact distance is measured how often do they encounter difficulties during the pin printing itself with bent or wrinkled grids?

We removed the sentence "as EM grids .. using the camera" as this was unclear to the referee.

We do not wish to use bent or wrinkled grids. In our workflow, grids are pre-mounted into the autogrid carrier at room temperature, which greatly reduces the risks of bending. The quality of the autogrid assembly can be assessed prior to sample deposition.

Even then, EM grids are still not perfectly flat over the whole area. Therefore, the location of the deposition area is determined visually by focussing. Focussing

takes place using a camera with an estimated accuracy of 1-2 micron and takes 30s. Since a standoff distance of around 10 micrometer is typically used, the pin never touches the grid avoiding potential damage to the grid.

To better explain this procedure, the end of the first paragraph of the **Pin Printing** section now reads "In order to determine the relative positions of the pin and the grid, we position them subsequently in the focal plane of the camera. Using these calibrated positions, the pin can be moved to a defined distance of the grid."

6. The speed of movement during pin printing is stated to be 0.3 mm/s. So the entire course of the pin printing likely takes on the order of several to many seconds? These numbers should be provided.

The entire course of pin printing (0.3mm/s, square spiral) takes 3s, as shown in figure 3b. It was also stated in the main text (section **Integration**).

The authors also completely ignore this time course in their subsequent discussion on the effects of interactions with the air water interface. They discuss the time lapse from the end of the pin printing to the subsequent vitrification (80ms) **as if this were the critical time lapse** whereas this is a small fraction of the total time the sample has dwelt in a thin liquid film due to the slow pin printing procedure.

We are fully aware of the effects of interactions with the air water interface and now made this explicit in the discussion. We believe that a critical time lapse should be much less than 1ms in order to outrun particle-surface interactions, and certainly did not want to claim the VitroJet could outrun it. The new text in the **Discussion** reads "Proteins tend to adsorb to the air-water interface where they can denature^{12, 33, 34, 35}. It seems intuitive that reducing the time the sample is exposed to such an interface would help to prevent protein denaturation¹⁷. However, using the Stokes Einstein equation, it has been calculated that even for a minimal residence time of ~ 1 ms, particle-surface interactions will occur dozens of times before the water is frozen^{35, 36}. Unfortunately, all existing devices have residence times varying between 11 ms²⁶ and several seconds and therefore cannot prevent particle-surface interactions. For the VitroJet, the residence time varies between seconds for the first written squares down to 80ms for the last square written prior to vitrification."

They also fail to cite or discuss a paper (Noble et al. Reducing effects of particle adsorption to the air-water interface in cryo-EM. Nature Methods) that proved that time courses on the order of 100ms were indeed useful for reducing interactions at the air-water interface.

We had already included reference to an Elife paper of Noble which discussed particle behaviour at the air-water interface. We have now also added the Nature Methods reference.

7. The authors state: "The initially deposited film layer could differ in thickness from the final one that is vitrified" and then mention possible causes: "evaporation, condensation, and liquid flow". They immediately discard the first two as being unlikely due to the dew point control and then continue on to a somewhat complicated argument as to why the third is also not possible based on their measurements. This leaves unanswered the question as to why the ice thickness changes. And once again emphasizes the critical need for more quantitative measurements of these aspects, as commented already above.

We agree with the reviewer that this paragraph was not clear enough and rewrote it entirely (see manuscript, end of **Pin Printing** section). It now also includes a reference to a new supplementary figure.

8. One of the critical steps in this method is jet vitrification, and the authors do not provide much detail. Flushing through a lot of ethane has implications in terms of costs and safety issues and these should be discussed.

The VitroJet ethane cup is filled with 13ml of liquid ethane, compared to 6-7ml in the Vitrobot. The jet itself only takes 50 milliseconds using a fraction of the total volume. The ethane from the jet is recycled for upcoming grids. Besides that, the cryogenic module is confined such that the user is not at risk. This has now been added to the manuscript as well, within the **Vitrification Module** section.

Additional comments:

9. In the Introduction, I think instead of "absorb" the mean to say "adsorb"

This has been corrected

10. The authors should mention the new TTP Labtech device, Chameleon, based on the "Spotiton" technology, when discussing commercial options in the introduction.

A reference to Chameleon (Darrow et al, 2019) has now been included in the **Discussion**

11. The only image of the overall instrument is insufficient and it is rather challenging to imagine the different sections of the instrument. Perhaps including a block diagram and some magnified images of the different parts of the device would help.

We now included some magnified images of the different parts of the device within figure 3a and updated the block diagram of figure 3b.

12. In Figure 3b, the time taken for making each grid is 3 mins. Is there time involved in refilling the ethane? Is this automated?

The ethane is recycled. We typically refill it after ca 8 grids, which is currently done manually and takes approximately 2 minutes. This will be automated in the next version of the VitroJet.

13. Can the gas mixtures of the glow discharge unit be varied and modified?

Within the described experiments, we only used ambient air. Indeed, it would be good to allow for different controlled gas mixtures in the glow discharge unit. The upcoming version of the VitroJet can accept different gas mixtures for the glow discharge unit, such as air, O₂, Argon, artificial air.

14. Why are only 16 squares used? How many holes does this include? How many of those holes are useable? To be useful these numbers should be included for all four samples so that reproducibility can be evaluated. I assume that only a single grid was used for each test sample but this should be clearly stated.

The field of view of the camera is 16 squares of a 300 mesh grid. It is possible to extend the deposition outside of the field of view, but this was not necessary in our case. Only one grid was used for each test sample, where we only used a number of squares. The number of squares and holes used per sample have now been included in the Methods section. The number of holes selected for data collection was not necessarily the maximum number we could use (beamtime limitations). Typical ice thickness distribution on a grid is shown as an answer to

point 4 and in the supplementary materials. The upcoming version of the VitroJet will have a camera with a significant larger field of view, which allows to monitor a larger deposition area.

15. They state: "Holes close to the grid bar were skipped to avoid thicker ice due to the wicking of the grid bars". Perhaps I just missed this in the complex argument about liquid flow but it seems holes next to the grids bars should be thinner not thicker if liquid close to the grids bars is wicked away?

Capillary forces in the corner between foil and mesh attract sample from the square. The layer that is very close to the hydrophilic grid bar is very thick due to the accumulation of sample when the pin moves over the square. However, capillary forces continue to deplete sample from the thin layer inside the square after deposition. In principle, this could become apparent indeed as a thinner ring inside the square. However, liquid flow is proportional to the thickness of the sample layer. Because we work with sub 100 nm thin layers, liquid flow in such layers becomes negligible at the time scales we used for pin printing.

We note that the paragraph on liquid flow has been significantly rewritten.

16. In Figure 3b, the time taken for making each grid is 3 mins, and it is showed that it is the same for the next grids. How about the time taken to refill ethane? Is it auto-filled? If yes, the authors should include that in this figure.

Within the current VitroJet, the ethane is filled once and then recycled, thus no need for auto-filling. We typically refill it (manually) after 8 grids, which takes approximately 2 minutes. A short paragraph has been added on the cryogen preparation within the **Vitrification Module** section.

Reviewer #2 (Remarks to the Author):

This manuscript describes novel and interesting methodological developments to automate cryoEM specimen preparation and make it more reproducible with less operator dependence. In order to accomplish this goal, the authors have created an integrated grid freezing device that includes a glow discharge module, a pin printing sample application module, and jet vitrification module, all of which operate on an already clipped autogrid. The manuscript is an interesting and valuable addition to the literature and should be published with minor revisions.

1. The authors have incorporated a feedback loop to keep the grids at dewpoint temperature – This objective is accomplished by visual inspection of the evaporation/condensation rate on the grid coupled with thermoelectric cooling/heating of the grid. More detail of how the feedback camera and software work would be helpful and interesting for the reader. Is there evidence that the entire feedback loop (visualization, electronics, heating/cooling) can operate on the same timescale as the evaporation process?

Some more details on how the feedback camera and software work are now included in the main text (section Process Chamber).

We now also included a real-time movie of how the feedback loop can be adjusted to induce evaporation and condensation (supplementary movie 1). Within a time lapse of 2 ½ minutes, the set grid temperature was altered three times, in steps of 0.5K. We used a dry grid to start with, and demonstrate that we can induce condensation when we cool the grid below the dewpoint temperature, and evaporation when the grid was heated above the dewpoint temperature. Albeit the condensation and evaporation seem to be slow in this movie, its effect on the layer thickness is dramatic as judged by the (dis)appearance of Newton rings. The grid is maintained at a fixed offset above and below dewpoint in this movie: the grid temperature can be set within a few hundredths of degrees. We believe that this would provide an advantage compared to the current scheme where no dewpoint controls are used at all.

2. Have the authors tested this device with different types of grids (continuous carbon, graphene oxide, holey carbon, different hole sizes) or with different buffers (detergents, glycerol, etc.)? In the discussion, the authors state that 'the printing procedure is compatible with a multitude of grid supports' and it seems it should work with different types of grids.

Correct. This sentence now reads "We tested the device with various types of grids (continuous carbon, graphene oxide, different hole sizes, UltrAuFoil) and found the pin printing procedure to be compatible with a multitude of (modified) grid supports." The samples described in the manuscript have relative similar buffer composition, but indeed, different constituents (detergents, glycerol) have also been successfully tested.

However, the claim that pin printing on graphene should avoid air-water interface interactions does not seem correct: with standard graphene oxide preparation the protein particle adsorbs to the surface while still in bulk solution, thereby avoiding the situation where the protein particle is in a thin film where it can encounter the air-water interface. With the pin printing method the protein would have to adsorb to the graphene after the thin film has already been created. Therefore, in this situation the protein could still encounter the air-water interface before adsorbing.

The volume between the pin and the sample carrier can still be seen as "bulk solvent" compared to the deposited layer. Given the stand-off distance of,

typically, 10 μ m, the protein could already adsorb to the graphene oxide prior to the creation of the <100nm thin film and encountering the air-water interface. The speed at which this volume moves over the surface, can be adjusted by the user. We agree with the reviewer that once the thin film is formed, protein molecules that have not been adsorbed yet onto the graphene film, could still encounter the air-water interface. This has now been better addressed in the discussion.

In this section the authors state that the air-water interface is referred to as the "the deadly touch". First, this sentence doesn't make sense (how can an interface be a touch?). Second, the phrase is certainly not common.

The reference to "the deadly touch" biorxiv manuscript has now been updated together with the new title of the accepted paper in Elife.

Also, we believe the first calculation of the frequency with which a protein in a thin film solution will encounter and airwater was due to Glaeser (e.g. Taylor, K.A., and Glaeser, R.M. (2008). J. Struct. Biol. 163, 214–223)

We certainly should have given credits to the authors who published the first calculations and now included this reference.

3. One of the major stated advantages of this instrument is its reproducibility/consistency. The authors state that 'From the squares that were pin-printed, most holes could be selected for data collection'. However, it seems like the authors are in a position to include a more quantitative assessment of the grids they produce (ie. number of good holes, ice thickness, etc per holes that are pin-printed). This kind of assessment can be useful – especially as other grid freezing devices are introduced to the market. Furthermore, does this reproducibility vary under different conditions (see question 2).
See also point 4 of reviewer 1, in which we show the reproducibility over the four grids. Statistics about the number of good holes have now been included in the main text. We like to refer to the prelude of this document concerning the extent of our statement and claims.
4. The authors state that the layer thickness can be tuned by altering the deposition parameters, such as the pin speed. This point is extremely important. Is there experimental evidence that the deposition thickness is related to how thick the ice is in the holes after vitrification? If so, this can be very powerful as ice thickness is a major factor when trying to get a sample to high resolution. It would be nice to demonstrate that ice thickness can indeed be controlled by the user.
Although other factors play a role with pin-printing, this process has similarities with dip-coating. With dip-coating, a substrate is retracted from a bath with liquid coating and it is well known that the thickness can be tuned using the velocity (David Quere, FLUID COATING ON A FIBER, Annu. Rev. Fluid Mech. 1999). In the current setup, the user can **tune** the layer thickness, however, a full **control** would require an even better understanding. We are working on a more technical follow-up manuscript, which discusses the theory and validation of the pinprinting technology. In there, we work with a simplified setup and a support incompatible with cryo-EM.